# Prevalence and Risk Factors of Bullying and Sexual and Racial Harassment in Healthcare Workers: A Cross-Sectional Study in Italy

**DOI:** 10.3390/ijerph19116938

**Published:** 2022-06-06

**Authors:** Giuseppe La Torre, Alberto Firenze, Corrado Colaprico, Eleonora Ricci, Luciano Pio Di Gioia, Dorotea Serò, Giuseppe Perri, Manuela Soncin, Dario Cremonesi, Nadia De Camillis, Sara Guidolin, Giulia Evangelista, Mattia Marte, Nicola Giovanni Fedele, Simone De Sio, Alice Mannocci, Sabina Sernia, Silvio Brusaferro

**Affiliations:** 1Department of Public Health and Infectious Diseases, Sapienza University of Rome, 00185 Rome, Italy; corrado.colaprico@uniroma1.it (C.C.); eleonora.ricci@uniroma1.it (E.R.); digioia.1546571@studenti.uniroma1.it (L.P.D.G.); sero.1755070@studenti.uniroma1.it (D.S.); giulia.evang@gmail.com (G.E.); mattia.marte@uniroma1.it (M.M.); fedele.1558181@studenti.uniroma1.it (N.G.F.); sabina.sernia@uniroma1.it (S.S.); 2Department of Health Promotion, Mother and Child Care, Internal Medicine and Medical Specialties, University of Palermo, 90133 Palermo, Italy; alberto.firenze@unipa.it; 3Dipartimento di Area Medica, Università di Udine, 33100 Udine, Italy; giuseppe.perri@uniud.it (G.P.); silvio.brusferro@uniud.it (S.B.); 4Ospedale Niguarda, 20162 Milano, Italy; mnl.mso@gmail.com; 5Ordine delle Professioni Infermieristiche, 22100 Como, Italy; info@opicomo.it; 6Local Health Unit ASL 2, 66100 Chieti, Italy; qualita@asl2abruzzo.it; 7University of Padua, 35122 Padua, Italy; sara.guidolin.2@studenti.unipd.it; 8R.U. of Occupational Medicine, Sapienza University of Rome, 00185 Rome, Italy; simone.desio@uniroma1.it; 9Universitas Mercatorum, 00186 Rome, Italy; alice.mannocci@unimercatorum.it

**Keywords:** bullying, sexual harassment, racial harassment, healthcare workers, cross-sectional, Italy

## Abstract

Background: This cross-sectional study aims to evaluate the prevalence and socio-demographic factors associated with workplace bullying, sexual harassment and racial harassment among Italian health workers. Methods: We recruited 3129 participants using an online Italian translation of the ‘Workplace Violence in the Health Sector Country Case Studies Research Instruments Survey’ (WVHS) questionnaire. Data were analyzed with univariate (chi-square) and multivariate (multiple logistic regression) analysis. Results: Univariate analysis shows that females are significantly more affected by bullying (16.4% vs. 12.3%) and sexual harassment (2.4% vs. 1.3%). On the other hand, males are significantly more affected by racial harassment (3.1% vs. 2.0%). Multivariate analysis shows higher odds of being affected by bullying (OR = 1.30; 95% CI (1.03, 1.64)) and sexual harassment (OR = 2.08; 95% CI (1.04, 4.00)) for females, and higher odds of undergoing racial harassment (OR = 1.55; 95% CI (0.95, 2.53)) for males. Conclusion: This analysis of work situations looks to identify those risk factors, existing or potential, that increase the probability of episodes of violence. A group of work or other subjects identified by direction will have to evaluate the vulnerability of workplaces and establish more effective preventive actions to be adopted.

## 1. Introduction

Workplace bullying is related to repeated actions and practices directed against one or more workers that are unwanted by the victim, may be carried out deliberately or unconsciously, clearly cause humiliation, offence and distress, and may interfere with job performance and/or cause an unpleasant working environment [1]. It has very serious consequences on health as it can lead to psychic disablement and occupational diseases, depression and psychosomatic symptoms. European research on this issue started in Scandinavia in the 1980s and spread during the late 1990s in many other European countries as well as Australia [2]. For one of the leading scientists in the field of workplace bullying, Leymann (who introduced the word “mobbing” into the scientific vocabulary), “psychological terror or mobbing in working life involves hostile and unethical communication which is directed in a systematic manner by one or more individuals, mainly toward one individual, who, due to mobbing, is pushed into a helpless and defenceless position and held there by means of continuing mobbing activities. These actions occur on a very frequent basis (at least once a week) and over a long period of time (at least six months duration)” [3].

On the other hand, the verb “to harass” is defined as “to disturb or irritate persistently”, “wear out, exhaust” or “ to enervate (an enemy) by repeated attacks or raids” [4]. Sexual harassment is defined by the European Commission as “conduct of a sexual nature, or other conduct based on sex affecting the dignity of women and men at work […] if: such conduct is unwanted, unreasonable and offensive to the recipient; a person’s rejection of, or submission to, such conduct on the part of employers or workers (including superiors or colleagues) is used explicitly or implicitly as a basis for a decision which affects that person’s access to vocational training, access to employment, continued employment, promotion or salary or any other employment decisions; such conduct creates an intimidating, hostile or humiliating work environment for the recipient” [5]. Experiences of sexually harassing behaviour are potential sources of stress for workers. Such negative behaviors undermine the physical and mental health of workers and reduce their productivity [6]. Racial harassment is an incident or a series of incidents intended or likely to intimidate, offend or harm an individual or group because of their ethnic origin, color, race or nationality, and a racist incident is any incident that is perceived to be racist by the victim or any other person. Such behaviors may include: derogatory name-calling, verbal threats, insults and racist jokes, displays of racially offensive material, exclusion from normal workplace conversation or activities, physical attacks, and encouraging others to commit any such acts [7]. Psychosocial research on mobbing is currently being carried out in a number of countries, mainly in Europe [8]. In an empirical study, Ege shows that post-traumatic embitterment disorder (PTED) is the most appropriate psychological diagnosis for victims of workplace conflicts, particularly bullying [9].

Since such studies are not frequent in our country, this study aimed to evaluate the epidemiology of bullying and sexual and racial harassment in Italian health workers in terms of:−The occurrence (prevalence) of the three outcomes;−The socio-economic factors associated with these outcomes.

## 2. Materials and Methods

### 2.1. Study Design

A cross-sectional study was carried out. This was a multicenter study that estimated the occurrence (prevalence) of bullying and sexual and racial harassment in Italian health workers and associated social-demographic factors. An Italian translation of the Workplace Violence in the Health Sector Country Case Studies Research Instruments Survey (WVHS) questionnaire was made. Questions were on emotional impact and work discomfort after an event of workplace violence. WVHS is structured in five sections: general personal criteria, workplace; a section about frequency and episodes of “physical violence” in the past 12 months in the workplace; a section about the presence of forms of “psychological violence” in the past 12 months divided into subsections (verbal abuse, Mobbing/Bullying, racial and sexual harassments); and a section about the adoption of policies of prevention/containment; personal opinions.

The questionnaire was then administered to the included healthcare professionals to prove the comprehensibility of the questions.

The questionnaire was administered as a digitized self-compilation rather than a direct interview to involve interviewees in a less confidential but more discretionary relationship because the questions proposed deal with fairly private and sensitive themes, which could somehow alter the truthfulness of their answers.

After that, the test was changed into an online questionnaire using a specific method of Google Docs.

Both healthcare professionals (doctors and nurses) and medical and nursing students were invited to complete the questionnaire to validate the Italian version.

### 2.2. Setting

The cross-sectional study was carried out in several centers that have voluntarily joined. In the subdivision of the sample by macro areas, the following criteria were adopted:−Northern Regions: Aosta Valley, Piedmont, Lumbardy, Trentino-Bolzano, Veneto, Friuli Venezia Giulia;−Southern-Central Regions: Liguria, Emilia Romagna, Tuscany, Marche, Umbria, Lazio, Molise, Apulia, Campania, Basilicata, Calabria, Sardinia, Sicily.

### 2.3. Participants

A total of 3129 subjects answered the questionnaire validly. The health figures included in this study were: nurses, doctors, other healthcare workers and trainee students. Women were 72.3% of all respondents. The most frequent age group was between 40 and 54 years. The most represented professional category is that of nurses, with 49.8% of responses. Most respondents have over 20 years of experience (48.2%).

### 2.4. Variables

The characteristics of the sample population that was considered were: gender, age, macro area, profession, and years of work. After that, we analyzed mobbing and sexual and racial harassment against health workers in the last 12 months. The features of the sample are described in Table 1.

### 2.5. Data Sources/Measurement

The following variables were considered:−Dependent variables: being a victim of bullying in the last 12 months (yes vs. no); being a victim of sexual harassment in the last 12 months (yes vs. no); being a victim of racial harassment in the last 12 months (yes vs. no);−Independent variables: gender, age, macro area, profession, and years of work.

### 2.6. Quantitative Variables

Quantitative variables (age and year of work) were transformed into categorical (ordinal) variables (three modalities). For the multivariate analysis, for each variable, two dummy variables were created.

### 2.7. Statistical Methods—Bias—Study Size

The collected data then were analysed with univariate analysis, which used the Chi-square test, and with multivariate analysis, which used models of multiple logistic regression.

The dependent variables of the three models were: (a) being a victim of bullying; (b) being a victim of sexual harassment; (c) being a victim of racial harassment.

The independent variables considered in the multivariate analysis were: gender, age, macro area, profession, and years of work.

The logistic regression models were built with the use of a full model and a stepwise model with a backward elimination procedure.

Results are presented as odds ratio (OR) and 95% confidence intervals (95% CI).

The statistical significance was set at *p* < 0.05.

Statistical analysis was carried out using SPSS, release 25.0.

## 3. Results

### 3.1. Univariate and Multivariate Analysis

#### 3.1.1. Being a Victim of Bullying in the Last 12 Months

The univariate analysis (Table 2) shows that the female gender is significantly more affected (16.4%), and the most affected age group is that between 35 and 39 years. There is a significant difference between regions: Northern ones (17.0%) have a greater presence of the phenomenon of bullying. There are no differences related to the exercised profession.

Multivariate analysis confirms that women are more exposed to bullying than men, and the male gender is a protective factor related to bullying (OR = 0.75; 95% CI (0.59, 0.94)): in the Northern regions of Italy, there is an increased risk of bullying (OR = 1.45; 95% CI (1.17, 1.80)).

#### 3.1.2. Sexual Harassment in the Past 12 Months

Univariate analysis (Table 3) shows that the female gender is significantly more affected (2.4%), and the most affected age group is between 20 and 24 years. For the age group over 50 years, there is a substantial decrease in people who have experienced sexual harassment. There are no differences related to the regions they belong to. There are no differences related to the exercised profession.

The multivariate analysis confirms that women are more exposed to sexual harassment than men, and the male gender is a protective factor related to sexual harassment (OR = 0.50; 95% CI (0.26, 0.97)); people over 20 years are in the working age most at risk (OR = 0.63; 95% CI (0.38, 1.05)).

#### 3.1.3. Racial Harassment in the Past 12 Months

Univariate analysis (Table 4) shows that the male gender is significantly more affected (3.1%), and the most affected age group is between 30 and 34 years. For the age group over 50 years, there is a substantial decrease in people who have experienced racial harassment. There is an important difference among regions: the Northern ones (2.6%) have a greater presence of this phenomenon. Nurses are more affected than doctors and other healthcare professionals.

Multivariate analysis confirms that men are more exposed to racial harassment then women (OR = 1.55; 95% CI (0.95, 2.53)). Nurses have a much higher risk of suffering racial harassment (OR = 2.46; IC 95% (1.51, 4.02)), and people who worked more then 11 years are more protected (OR = 0.27; IC 95% (0.15, 0.51)).

## 4. Discussion

The study shows that bullying stands at 15.3%, sexual harassment stands at 2.1%, and racial harassment stands at 2.3% of the sample in the last 12 months. This data is in line with studies conducted on Italian and other nationality healthcare professionals.

The female gender, as in other studies, is more affected by bullying, at 16.4%, and by sexual harassment, at 2.4%, while as regards racial harassment, the male gender is the most affected, with 3.1% of the sample. Nurses are the most affected category by harassment in the last 12 months. This data is in line with the trend registered in Italy. There are no significant differences among healthcare professionals from different macroareas.

This study also reveals how the problem of aggression against the healthcare workers is still a central problem, and it provides an additional cue for research in the analysis of the phenomenon. In particular, some aspects of the frequency of aggressions in the relative reactions are evident in association with belonging to macro areas of victims, which deserves further investigation.

The study has, among its strong points, a discrete sample with 3129 participants, which comprises one of the largest studies ever carried out in Italy on the prevalence and socio-demographic factors associated with workplace bullying and sexual and racial harassment in the healthcare sector. The limits of this study include the voluntary involvement of some structures that may have approached this study because they could be sensible to the problem, which could generate a selection bias. In addition, for the same assumption, there is a not completed representativeness of some regions because the involvement was not homogenous.

All the aspects highlighted in the analysis of the obtained results still show that there is a long way to go in the prevention of aggression to healthcare workers and in supporting them.

### 4.1. Bullying

In our sample, almost one-sixth of the participants reported being victims of workplace bullying in the last 12 months in the healthcare sector. This figure must be considered with attention since there is often an association between bullying and work-related stress. De Sio et al., in a cross-sectional study with 191 participants about bullying and work-related stress in healthcare workers, pointed out that exposure to bullying can influence the perception of psychosocial risks. The workers were divided into two groups: those who were exposed to bullying at work and those who were not. After that, the authors administered to the workers the HSE questionnaire, which was aimed at assessing the presence of work-related stress, and, finally, they performed the statistical analysis of the data: the difference between the high-exposure group and the low-exposure group was statistically significant [10].

Females and nurses are more affected by bullying behaviors, and this is in line with other studies carried out at the international level. A cross-sectional study by Njaka et al. was conducted in a clinic in Barbados of 141 healthcare workers. The results indicated that female staff and nurses experienced more abuse than males and physicians. However, this was a small sample size study [11]. Voltz et al. investigated the horizontal violence (HV), referring to bullying, verbal or physical threats, emotional abuse, and other purposeful malicious acts, in a sample of 91 healthcare workers among physicians, residents, and MLPs. The result of this study showed that the prevalence of reported behaviors ranged from 1.1% (physical assault) to 34.1% (been shouted at). This study is in line with our result regarding mobbing (15.3%) [12]. Moreover, a higher prevalence of workplace bullying was reported by Ariza-Montes et al., who, in their research paper, identified the determinants of workplace bullying among healthcare professionals that emerge from personal variables, working conditions, and contextual factors. Workplace bullying emerges as even more acute among female healthcare workers (72.6% compared to 52.7%) and young workers (46.2% compared to 27.3% between 25 and 39 years old) [13].

The occurrence of bullying behaviors in the healthcare sector also has practical consequences since there is an association between this issue with job satisfaction, as well as psychological stress, sleep quality and subjective health. In their study, Erdogan and Yildrim aimed to determine healthcare professionals’ exposure to mobbing behaviors and the relation of mobbing with job satisfaction and organizational commitment in 479 healthcare professionals. Females, as compared with males, and participants with low income, as compared to high income, were more exposed to mobbing, and this is in line with our study. Other results that we also found in our study include, first of all, that nurses, as compared with doctors, were more exposed to mobbing and that individuals with an occupational experience of >10 years were more exposed to mobbing [14]. Sun et al. identified the incidence rate of workplace violence, including mobbing, in a cross-selectional online survey study with a total of 3016 participants. The results demonstrated, in a higher percentage than our study, that the prevalence rate of mobbing behaviour was 40.2%. The results also demonstrated the association with psychological stress, sleep quality and subjective health [15]. Another cross-sectional study by Dirican et al. was conducted on 752 medical staff in primary health care units. The result was that the midwife-nurse group was the largest group reporting being bullied. Moreover, the target period of mobbing was mostly longer than 18 months [16].

### 4.2. Harassment

Our data revealed that one-fiftieth of the participants reported being a victim of workplace sexual and racial harassment in the last 12 months in the healthcare sector. These results are much lower than the reported frequency of these outcomes in other international studies. In their paper, Boafo et al. documented the incidence and the effect of workplace verbal abuse and sexual harassment against Ghanaian nurses. This article was a cross-sectional study with 592 professional nurses from five different hospitals. A total of 72 (12.2%) nurses reported that they had been sexually harassed in their workplace in the past 12 months before the study, 50% of which occurred by a medical doctor. The study highlighted that there was a difference between gender, with 55.5% of females and 43.1% of males referring to verbal abuse. This finding is consistent with a previous study [17].

However, being a female and a nurse is confirmed to have higher odds of being a victim of these harassments. Kahsay et al. undertook a systematic review to determine the prevalence of sexual harassment against female nurses, as well as the types, perpetrators, and health consequences of the harassment. The prevalence of sexual harassment against female nurses was 43.15%, and this is in line with the result of our study but in a higher percentage. Notably, 46.59% of them were harassed by patients, 41.10% by physicians, 27.74% by patients’ family, 20% by nurses, and 17.8% by other coworker perpetrators [18]. In a study by Shafran-Tikva et al., a quantitative questionnaire was administered to 729 physicians and nurses in a large general hospital, focusing on verbal violence and sexual harassment. The result was that nurses were exposed to violence almost twice as much as physicians. This result is different from our study since doctors appear to be a little more exposed to sexual abuse than nurses [19]. Rhead et al. examined the impact of harassment and discrimination on National Health Service (NHS) staff working in London trusts, utilizing data from the 2019 TIDES cross-sectional survey. In total, 931 London-based healthcare practitioners participated in the TIDES survey. They showed that women, black ethnic minority staff and migrants were most at risk of discrimination and/or harassment. In addition, as in our study, nurses were more exposed to racial harassment than doctors [20]. Finally, Shields and Price investigate the determinants of perceived racial harassment in the workplace and its impact on job satisfaction and quitting behaviour among ethnic minority nurses using data from a unique large-scale survey of British NHS nurses. Nearly 40% of ethnic minority nurses report experiencing racial harassment from work colleagues, while more than 64% report suffering racial harassment from patients. Such racial harassment is found to lead to a significant reduction in job satisfaction, which, in turn, increases nurses’ intentions to quit their job [21].

### 4.3. How to Prevent and Manage

The prevention of violent acts against workers requires that health organizations identify risk factors for personnel safety and apply the most appropriate strategies. In this way, health structures must implement a prevention program of violence.

The constitution of a work group could be useful to encourage the involvement of a management team and for personnel more exposed to the risk in order to allow individuation and implementation of actions and necessary measures to ensure the effectiveness of the program. We need to recognize that reporting and addressing workplace harassment and bullying behaviours is highly important. There are several implications for healthcare institution management. Issues such as acknowledgement of the problem, as well as transformational leadership and prioritization of psychosocial issues are all elements to consider when tackling these behaviours [22].

However, this is not sufficient, since healthcare institutions need to put in place policies that clearly underline the unacceptability of these behaviours. Moreover, these institutions need to follow a systematic method to comprehensively investigate allegations of racial and sexual harassment, as well as workplace bullying, in order to actively make determinations as to whether allegations are substantiated [23].

The implementation of practical issues is often lacking. Health care providers must establish a process for managing disruptive behaviours [24].

An appropriate investigation is mandatory, as is protection for those who report abuses. There must be space for a system, forums and safe space for open communication between employers and healthcare workers for follow-up regarding harassment or bullying within the healthcare organization [25,26].

## 5. Conclusions

During their work, health workers of hospitals and local structures are exposed to several factors which could be dangerous to health and security. A relevant risk is to live a violent experience such as aggression, harassment or other criminal events involving personal injury.

The aim of hospitals must be to prevent acts of violence against workers using measures of elimination or reduction of present risk and acquisition of skills by workers in order to manage and evaluate these events when these happen.

## Figures and Tables

**Table 1 ijerph-19-06938-t001:** Characteristics of the sample population.

Variable	*N* (%)
Gender	
Female	2261 (72.3%)
Male	868 (27.7%)
Age group (years)	
<40	902 (28.8%)
40–54	1403 (44.9%)
≥55	823 (26.3%)
Missing value = 1	
Macro area	
Northern	2029 (64.8%)
Southern-Central	1100 (35.2%)
Profession	
Doctor	710 (22.7%)
Nurse	1557 (49.8%)
Other healthcare professions	862 (27.5%)
Years of work	
≤10	733 (23.4%)
20-Nov	888 (28.4%)
>20	1508 (48.2%)
Have you been bullying in the last 12 months?	
No	2651 (84.7%)
Yes	478 (15.3%)
Have you been sexually harassed in the last 12 months?	
No	3063 (97.9%)
Yes	66 (2.1%)
Have you been racially harassed in the last 12 months?	
No	3056 (97.7%)
Yes	73 (2.3%)

**Table 2 ijerph-19-06938-t002:** Being a victim of bullying: univariate and multivariate analysis.

Variable	Bullying			
	Univariate Analysis		Multivariate Analysis	
	No *n* (%)	Yes *n* (%)	Full Model—Step 1 OR (95% CI)	Stepwise Model
				OR (95% CI)
Gender				
Female	1890 (83.6%)	371 (16.4%)	1.30 (1.03, 1.64)	1.33 (1.06, 1.70)
Male	761 (87.7%)	107 (12.3%)	1	1
Age group (years)				
<40	756 (83.8%)	146 (16.2%)	1	
40–54	1183 (84.3%)	220 (15.7%)	0.90 (0.64, 1.25)	
≥55	711 (86.4%)	112 (13.6%)	0.81 (0.54, 1.21)	
Missing value = 1				
Macro area				
North	1684 (83.0%)	345 (17.0%)	1.42 (1.13, 1.77)	1.45 (1.17, 1.80)
South-Center	967 (87.9%)	133 (12.1%)	1	1
Profession				
Doctor	608 (85.6%)	102 (14.4%)	0.92 (0.69, 1.22)	
Nurse	1317 (84.6%)	240 (15.4%)	1.06 (0.84, 1.35)	
Other healthcare professions	726 (84.2%)	136 (15.8%)	1	
Years of work				
≤10	626 (86.4%)	107 (14.6%)	1	
20-Nov	750 (84.6%)	138 (15.4%)	0.93 (0.69, 1.29)	
>20	1275 (84.5%)	233 (15.5%)	1.04 (0.71, 1.52)	

**Table 3 ijerph-19-06938-t003:** Being a victim of sexual harassment: univariate and multivariate analysis.

Variable	Sexual Harassment			
	Univariate Analysis		Multivariate Analysis	
	No *n* (%)	Yes *n* (%)	Full Model—Step 1 OR (IC 95%)	Stepwise Model
				OR (IC 95%)
Gender				
Female	2206 (97.6%)	55 (2.4%)	2.08 (1.08, 4.00)	1.96 (1.03, 3.84)
Male	877 (98.7%)	11 (1.3%)	1	1
Age group (years)				
<40	874 (96.9%)	28 (3.1%)	1	
40–54	1379 (98.3%)	24 (1.7%)	0.67 (0.31, 1.49)	
≥55	809 (98.3%)	14 (1.7%)	0.74 (0.28, 1.99)	
Missing value = 1				
Macro area				
North	1987 (97.9%)	42 (2.1%)	0.94 (0.56, 1.59)	
South-Center	1076 (97.8%)	24 (2.2%)	1	
Profession				
Doctor	692 (97.5%)	18 (2.5%)	1.37 (0.69, 2.73)	
Nurse	1524 (97.9%)	33 (2.1%)	1.20 (0.64, 2.24)	
Other healthcare professions	847 (98.3%)	15 (1.7%)	1	
Years of work				
≤10	715 (97.5%)	18 (2.5%)	1	1
20-Nov	871 (98.1%)	17 (1.9%)	0.77 (0.36, 1.62)	-
>20	1477 (97.9%)	31 (2.1%)	0.70 (0.27, 1.79)	0.64 (0.39, 1.05)

**Table 4 ijerph-19-06938-t004:** Being a victim of racial harassment: univariate and multivariate analysis.

Variable	Racial Harassments			
	Univariate Analysis		Multivariate Analysis	
	No *n* (%)	Yes *n* (%)	Full Model—Step 1 OR (IC 95%)	Stepwise Model
				OR (IC 95%)
Gender				
Female	2215 (98.0%)	46 (2.0%)	0.62 (0.38, 1.02)	0.64 (0.40, 1.05)
Male	841(96.9%)	27 (3.1%)	1	1
Age group (years)				
<40	862 (95.6%)	40 (4.4%)	1	
40–54	1377 (98.2%)	26 (1.8%)	0.99 (0.48, 2.08)	
≥55	816 (99.2%)	7 (0.8%)	0.64 (0.22, 1.93)	
Missing value = 1				
Macro area				
North	1977 (97.4%)	52 (2.6%)	1.38 (0.82, 2.35)	
South-Center	1079 (98.1%)	21 (1.9%)	1	
Profession				
Doctor	697 (98.2%)	13 (1.8%)	1.71 (0.53, 2.59)	-
Nurse	1509 (96.9%)	48 (3.1%)	2.56 (1.32, 4.94)	2.47 (1.51, 4.03)
Other healthcare professions	850 (98.6%)	12 (1.4%)	1	1
Years of work				
≤10	717 (95.6%)	16 (4.4%)	1	1
20-Nov	865 (97.4%)	23 (2.6%)	0.30 (0.15, 0.64)	0.28 (0.15, 0.51)
>20	1474 (97.7%)	34 (2.3%)	0.24 (0.10, 0.58)	0.21 (0.12, 0.36)

## Data Availability

Data are available upon request to the author for correspondence.

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
