# Peer review of "Prevalence and Risk Factors of Bullying and Sexual and Racial Harassment in Healthcare Workers: A Cross-Sectional Study in Italy"

_ijerph, 2022, doi:10.3390/ijerph19116938_

Round 1

Reviewer 1 Report

Thank you for your work. The authors report the results of an important and interesting study on workplace bullying and harassment among Italian healthcare professionals.

Despite its overall merits, I think the paper will improve considerably if the authors work more on the following areas:

1. It is unclear what the authors mean by "Epidemiology of Bullying, Sexual and Racial Harassments". It would be necessary for the authors to elaborate more on this in the Introduction and follow up in the Discussion that explains how they contribute to the relevant literature.

2. In the Discussion, the authors bring in many studies pertaining to workplace bullying and harassment. But it falls short of clarifying the logical linkages with the current study. The authors should discuss the significance of the current findings in light of the previous findings, instead of just listing several previous studies in this section.

3. The paper will be much easier to read and understand with some help from professional English editors.

Good luck with your revision and future research.

Reviewer 2 Report

I loved the article.  I have one question and one required change.  The question is:  Why are there no  hypothesis?  They should be included. This issue is why I checked - "Is the research design appropriate" - could be improved.

The required change is - on page 2 - line 63 - the authors list religion with racial harassment.  Religion is not racial.  It is not at all clear why it is included. Additionally, no where else in the document is religion mentioned.  The term needs to be deleted.

Reviewer 3 Report

The study is very interesting and has great applicative relevance in dealing with epidemiological data on mobbing; it is also theoretically and methodologically founded.

The statistical analyzes are conducted in an absolutely flawless way, however, the authors should find a way to explain the data, especially those relating to gender, since the psycho-social literature highlights that aggressive behaviours are mainly typical of the male gender. What factors can be useful to understand this data? New socialization to the genre? Greater precariousness and, therefore, a tendency to aggressive behaviour? A similar data was recently found in social media by Paciello et al (Paciello ,. (2021). Online sexist meme and its effects on moral and emotional processes in social media. Computers in human behavior, 116, 106655.) in which it emerges that in the face of sexist aggression, women appear to be more aggressive compared to men (more ironic ..). I would ask the authors to deepen these aspects.

Furthermore, I would suggest investigating the intervention side, namely that recent literature highlights that as virtual reality devices, subjects are much more likely to empathize with people in the difficulty of different ethnic backgrounds. Authors should further investigate and propose possible strategies.
